# CONCURRENT ADVERSARIAL LEARNING FOR LARGE-BATCH TRAINING

**Yong Liu[1], Xiangning Chen[2], Minhao Cheng[3], Cho-Jui Hsieh[2], Yang You[1]**
[1]Department of Computer Science, National University of Singapore
[2]Department of Computer Science, University of California, Los Angeles
[3]Department of Computer Science and Engineering, Hong Kong University of Science and Technology
{liuyong, youy}@comp.nus.edu.sg, {xiangning, chohsieh}@cs.ucla.edu,
minhaocheng@ust.hk

## ABSTRACT

Large-batch training has become a widely used technique when training neural networks with a large number of GPU/TPU processors. As batch size increases, stochastic optimizers tend to converge to sharp local minima, leading to degraded test performance. Current methods usually use extensive data augmentation to increase the batch size as a remedy, but we found the performance brought by data augmentation decreases with the increase of batch size. In this paper, we propose to leverage adversarial learning to increase the batch size in large-batch training. Despite being a natural choice for smoothing the decision surface and biasing towards a flat region, adversarial learning has not been successfully applied in large-batch training since it requires at least two sequential gradient computations at each step. To overcome this issue, we propose a novel Concurrent Adversarial Learning (ConAdv) method that decouples the sequential gradient computations in adversarial learning by utilizing stale parameters. Experimental results demonstrate that ConAdv can successfully increase the batch size of ResNet-50 training on ImageNet while maintaining high accuracy. This is the first work that successfully scales the ResNet-50 training batch size to 96K.

## 1 INTRODUCTION

As larger datasets and bigger models being proposed, training deep neural networks has become quite time-consuming. For instance, training BERT (Devlin et al., 2019) takes 3 days on 16 v3 TPUs. GPT-2 (Radford et al., 2019) contains 1,542M parameters and requires 168 hours of training on 16 v3 TPUs. With the developments of high performance computing clusters, (e.g., Google and NVIDIA build high performance clusters with thousands of TPU or GPU chips), how to fully utilize those computing resources to accelerate the training process has become an important research topic.

Data parallelism is a commonly used technique for distributed neural network training, where each processor computes the gradient of a local batch and the gradients across processors are aggregated at each iteration for a parameter update. Training with hundreds or thousands of processors with data parallelism is thus equivalent to running a stochastic gradient optimizer (e.g., SGD or Adam) with very large batch size, also known as **large batch training**. For example, Google and NVIDIA show that by increasing the batch size to 64k on ImageNet, they can finish 90-epochs ResNet training within one minute (Kumar et al., 2021; Mattson et al., 2019).

But why can't we infinitely increase the batch size as long as more computing resources are available? Large batch training often faces two challenges. First, given a fixed number of training epochs, increasing the batch size implies reducing the number of training iterations. Even worse, it has been observed that large-batch training often converges to solutions with bad generalization performance (also known as sharp local minima) (Keskar et al., 2017), possibly due to the lack of inherent noise in each stochastic gradient update. Although this problem can be partially mitigated by using different optimizers such as LARS (You et al., 2017) and LAMB (You et al., 2019), the limit of batch size still exists. For instance, Google utilizes several techniques, such as distributed batch normaliza-

tion and mixed-precision training, to further scale the training of ResNet-50 on 4096 v3 TPU chips. However, it can just expand the batch size to 64k (Kumar et al., 2021; Ying et al., 2018).

To mitigate the generalization gap brought by large-batch training, data augmentation has become an indispensable component. For instance, researchers at Facebook use augmentation to scale the training of ResNet-50 to 256 NVIDIA P100 GPUs with a batch size of 8k on ImageNet (Goyal et al., 2017). You et al. (2018) also use data augmentation to expand the batch size to 32k on 2048 KNL nodes. However, in this paper we observe that when batch size is large enough (i.e., larger than 32k), data augmentation will also increase the difficulty of training and even have a negative impact on test accuracy.

This motivates us to study the application of adversarial training in large-batch training, which finds a perturbation within a bounded set around each sample to train the model. Previous works find that adversarial training can lead to a significant decrease in the curvature of the loss surface and make the network behave more "linearly" in small batch size cases, which could be used as a way to improve generalization (Xie et al., 2020; Moosavi-Dezfooli et al., 2019). However, adversarial training has not been used in large-batch training since it requires a series of sequential gradient computations within each update to find an adversarial example. Even when conducting only one gradient ascent to find the adversarial example, adversarial training requires two sequential gradient computations (one for adversarial example and one for weight update) that cannot be parallelized. Therefore, even with infinite computing resources, adversarial training is at least two times slower than standard training and increasing the batch size cannot compensate for that.

To resolve this issue and make adversarial training applicable for large-batch training, we propose a novel Concurrent Adversarial Learning (ConAdv) algorithm. We show that by allowing the computation of adversarial examples using stale weights, the two sequential gradient computations in adversarial training can be decoupled, leading to fully parallelized computations at each step. As a result, extra processors can be fully utilized to achieve the same iteration throughput as original SGD or Adam optimizers. Comprehensive experimental results on large-batch training demonstrate that ConAdv is a better choice than existing augmentations.

Our main contributions are listed below:

- This is the first work showing that adversarial learning can significantly increase the batch size limit of large-batch training without using data augmentation.
- The proposed algorithm, ConvAdv, can successfully decouple the two sequential gradient computations in adversarial training and make them parallelizable. This makes adversarial training achieve similar efficiency with standard stochastic optimizers when using sufficient computing resources. Furthermore, we empirically show that ConAdv achieves almost identical performance as the original adversarial training. We also provide theoretical analysis for ConAdv.
- Comprehensive experimental studies demonstrate that the proposed method can push the limit of large batch training on various tasks. For ResNet-50 training on ImageNet, ConAdv alone achieves 75.3% accuracy when using 96K batch size. Further, the accuracy will rise to 76.2% when combined with data augmentation. This is the first method scaling ResNet-50 batch size to 96K with accuracy matching the MLPerf standard (75.9%), while previous methods fail to scale beyond 64K batch size.

## 2 BACKGROUND

### 2.1 LARGE-BATCH TRAINING

Using data parallelism with SGD naturally leads to large-batch training on distributed systems. However, it was shown that an extremely large batch is difficult to converge and has a generalization gap (Keskar et al., 2017; Hoffer et al., 2017) . Therefore, related work starts to carefully fine-tune the hyper-parameters to bridge the gap, such as learning rate, momentum (You et al., 2018; Goyal et al., 2017; Li, 2017; Shallue et al., 2018; Xue et al., 2021; Lou et al., 2021). Goyal et al. (2017) try to narrow the generalization gap with the heuristics of learning rate scaling. However, there is still big room to increase the batch size. Several recent works try to use adaptive learning rate to reduce the fine-tuning of hyper-parameters and further scaling the batch size to larger value (You et al., 2018; Iandola et al., 2016; Codreanu et al., 2017; Akiba et al., 2017; Jia et al., 2018; Smith

et al., 2017; Martens & Grosse, 2015; Devarakonda et al., 2017; Osawa et al., 2018; You et al., 2019; Yamazaki et al., 2019; Liu et al., 2022). You et al. (2017) propose Layer-wise Adaptive Rate Scaling (LARS) for better optimization and scaling to the batch size of 32k without performance penalty on ImageNet. In addition, related work also tries to bridge the gap from the aspect of augmentation. Goyal et al. (2017) use data augmentation to further scale the training of ResNet-50 on ImageNet. Yao et al. (2018a) propose an adaptive batch size method based on Hessian information to gradually increase batch size during training and use vanilla adversarial training to regularize against the sharp minima. However, the process of adversarial training is time-consuming and they just use the batch size of 16k in the second half of the training process (the initial batch size is 256). How to further accelerate the training process based on adversarial training and reduce its computational burden is still an open problem.

## 2.2 ADVERSARIAL LEARNING

Adversarial training has shown great success on improving the model robustness through collecting adversarial examples and injecting them into training data (Goodfellow et al., 2015; Papernot et al., 2016; Wang et al., 2019). Madry et al. (2017) formulates it into a min-max optimization framework as follows:

$$\min_{\theta} \mathbb{E}_{(x_i, y_i) \sim \mathbb{D}} \big[ \max_{||\delta||_p \in \epsilon} \mathcal{L}(\theta_t, x + \delta, y) \big], \tag{1}$$

where $\mathbb{D} = \{(x_i, y_i)\}_{i=1}^{n}$ denotes training samples and $x_i \in \mathbb{R}^d$, $y_i \in \{1, ..., Z\}$, $\delta$ is the adversarial perturbation, $|| \cdot ||_p$ denotes some $\mathcal{L}_p$-norm distance metric, $\theta_t$ is the parameters of time $t$ and Z is the number of classes. Goodfellow et al. (2015) proposes FGSM to collect adversarial data, which performs a one-step update along the gradient direction (the sign) of the loss function. Project Gradient Descent (PGD) algorithm (Madry et al., 2017) firstly carries out a random initial search in the allowable range (spherical noise region) near the original input, and then iterates FGSM several times to generate adversarial examples. Recently, several papers (Shafahi et al., 2019; Wong et al., 2020; Andriushchenko & Flammarion, 2020) aim to improve the computation overhead brought by adversarial training. Specifically, FreeAdv (Shafahi et al., 2019) tries to update both weight parameter $\theta$ and adversarial example $x$ at the same time by exploiting the correlation between the gradient to the input and to the model weights. Similar to Free-adv, Zhang et al. (2019) further restrict most of the forward and backpropagation within the first layer to speedup computation. Wong et al. (2020) finds the overhead could be further reduced by using single-step FGSM with random initialization. While these works aim to improve the efficiency of adversarial training, they still require at least two sequential gradient computations for every step. Our concurrent framework could decouple the two sequential gradient computations to further boost efficiency, which is more suitable for large-batch training. Recently, several works (Xie et al., 2020; Cheng et al., 2021; Chen et al., 2021; Mei et al., 2022) show that the adversarial example can serve as an augmentation to benefit the clean accuracy in the small batch size setting. However, whether adversarial training can improve the performance of large-batch training is still an open problem.

## 2.3 MLPERF

MLPerf (Mattson et al., 2019) is an industry-standard performance benchmark for machine learning, which aims to fairly evaluate system performance. Currently, it includes several representative tasks from major ML areas, such as vision, language, recommendation. In this paper, we use ResNet-50 (He et al., 2016) as our baseline model and the convergence baseline is 75.9% accuracy on ImageNet.

## 3 PROPOSED ALGORITHM

In this section, we introduce our enlightening findings and the proposed algorithm. We first study the limitation of data augmentation in large-batch training. Then we discuss the bottleneck of adversarial training in distributed systems and propose a novel Concurrent Adversarial Learning (ConAdv) method for large-batch training.

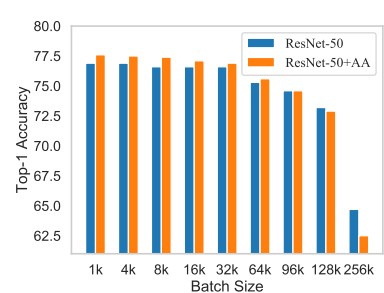

Figure 1: (a) Distributed Adversarial Learning (DisAdv), (b) Concurrent Adversarial Learning (ConAdv). To ease the understanding, we just show the system including two workers.

### 3.1 DOES DATA AUGMENTATION IMPROVE THE PERFORMANCE OF LARGE-BATCH TRAINING?

Data augmentation can usually improve the generalization of models and is a commonly used technique to improve the batch size limit in large-batch training. To formally study the effect of data augmentation in large-batch training, we train ResNet-50 using ImageNet (Deng et al., 2009) by AutoAug (AA) (Cubuk et al., 2019). The results shown in Figure 2 reveal that although AA helps improve generalization under batch size $\leq$ 64K, the performance gain decreases as batch size increases. Further, it could lead a negative effect when the batch size is large enough (e.g., 128K or 256K). For instance, the top-1 accuracy is increased from 76.9% to 77.5% when using AA on 1k batch size. However, it decreases from 73.2% to 72.9% under data augmentation when the batch size is 128k and drops from 64.7% to 62.5% when the batch size is 256k. The main reason is that the augmented data increases the diversity of training data, which leads to slower convergence when using fewer training iterations. Recent work tries to concat the original data and augmented data to jointly train the model and improve their accuracy (Berman et al., 2019). However, we find that concating them will hurt the accuracy when batch size is large. Therefore, we just use the augmented data to train the model. The above experimental results motivate us to explore a new method for large batch training.

### 3.2 ADVERSARIAL LEARNING IN THE DISTRIBUTED SETTING

Adversarial learning can be viewed as a way to automatically conduct data augmentation. Instead of defining fixed rules to augment data, adversarial learning conducts gradient-based adversarial attacks to find adversarial examples. As a result, adversarial learning leads to smoother decision boundary (Karimi et al., 2019; Madry et al., 2017), which often comes with flatter local minima (Yao et al., 2018b). Instead of solving the original empirical risk minimization problem, adversarial learning aims to solve a min-max objective that minimizes the loss under the worst case perturbation of samples within a small radius. In this paper, since our main goal is to improve clean accuracy instead of robustness, we consider the following training objective that includes loss on both natural samples and adversarial samples:

Figure 2: Augmentation Analysis

$$\min_{\theta} \mathbb{E}_{(x_i,y_i) \sim \mathbb{D}}[\mathcal{L}(\theta_t; x_i, y_i) + \max_{\|\delta\|_p \in \epsilon} \mathcal{L}(\theta_t; x_i + \delta, y_i)], \qquad (2)$$

where $\mathcal{L}$ is the loss function and $\epsilon$ represents the value of perturbation. Although many previous work in adversarial training focus on improving the trade-off between accuracy and robustness (Shafahi et al., 2019; Wong et al., 2020), recently Xie et al. (2020) show that using split BatchNorm for adversarial and clean data can improve the test performance on clean data. Therefore, we also adopt this split BatchNorm approach.

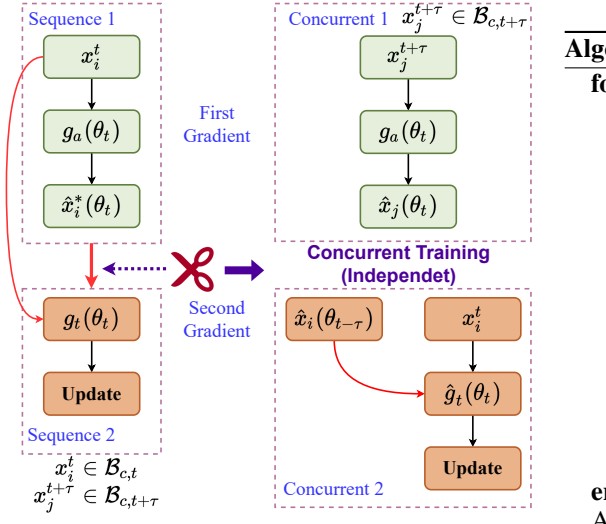

Figure 3: Vanilla Training and Concurrent Training

**Algorithm 1** ConAdv

> **for** $t = 1, \cdots, T$ **do**
>> **for** $x_i \in \mathcal{B}_{c,t}^k$ **do**
>>> Compute Loss:
>>> $\mathcal{L}(\theta_t; x_i, y_i)$ using main BN,
>>> $\mathcal{L}_a^k(\theta_t; \hat{x}_i(\theta_{t-\tau}), y_i)$ using adv BN,
>>> $\mathcal{L}_B(\theta_t) = \mathbb{E}_{\mathcal{B}_{c,t}^k} \mathcal{L}(\theta_t; x_i, y_i) +$
>>> $\qquad \mathbb{E}_{\mathcal{B}_{a,t}^k}(\hat{x}_i(\theta_{t-\tau}), y_i)$
>>> Minimize the $\mathcal{L}_B(\theta_t)$ and obtain $g_t^k(\theta_t)$
>> **end for**
>> **for** $x_i \in \mathcal{B}_{c,t+\tau}^k$ **do**
>>> Calculate adv gradient $g_a^k(\theta_t)$ on $\mathcal{B}_{c,t+\tau}^k$
>>> Obtain adv examples $(\hat{x}_i(\theta_t), y_i)$
>> **end for**
> **end for**
> Aggregate: $\hat{g}_t(\theta_t) = \frac{1}{K} \sum_{k=1}^K \hat{g}_t^k(\theta_t)$
> Update weight $\theta_{t+1}$ on parameter sever

For Distributed Adversarial Learning (DisAdv), training data $\mathbb{D}$ is partitioned into $N$ local dataset $\mathbb{D}^k$, and $\mathbb{D} = \cup_{k=1}^{k=K} \mathbb{D}^k$. For worker $k$, we firstly sample a mini-batch data (clean data) $\mathcal{B}_{t,c}^k$ from the local dataset $\mathbb{D}^k$ at each step $t$. After that, each worker downloads the weights $\theta_t$ from parameter sever and then uses $\theta_t$ to obtain the adversarial gradients $g_a^k(\theta_t) = \nabla_x \mathcal{L}(\theta_t; x_i, y_i)$ on input example $x_i \in \mathcal{B}_{t,c}^k$. Noted that we just use the local loss $\mathbb{E}_{(x_i, y_i) \sim \mathbb{D}^k} \mathcal{L}(\theta_t; x_i, y_i)$ to calculate the adversarial gradient $g_a^k(\theta_t)$ rather than the global loss $\mathbb{E}_{(x_i, y_i) \sim \mathbb{D}} \mathcal{L}(\theta_t; x_i, y_i)$, since we aim to reduce the communication cost between workers. In addition, we use 1-step Project Gradient Descent (PGD) to calculate $\hat{x}_i^*(\theta_t) = x_i + \alpha \cdot \nabla_x \mathcal{L}(\theta_t; x_i, y_i)$ to approximate the optimal adversarial example $x_i^*$. Therefore, we can collect the adversarial mini-batch $\mathcal{B}_{a,t}^k = \{(\hat{x}_i^*(\theta_t), y_i)\}$ and use both the clean example $(x_i, y_i) \in \mathcal{B}_{c,t}^k$ and adversarial example $(\hat{x}_i^*(\theta_t), y_i) \in \mathcal{B}_{a,t}^k$ to update the weights $\theta_t$. More specially, we use main BatchNorm to calculate the statics of clean data and auxiliary BatchNorm to obtain the statics of adversarial data.

We show the workflow of adversarial learning on distributed systems (DisAdv) as Figure 1, and more importantly, we notice that it requires two sequential gradient computations at each step which is time-consuming and, thus, not suitable for large-batch training. Specifically, we firstly need to compute the gradient $g_a^k(\theta_t)$ to collect adversarial example $\hat{x}^*$. After that, we use these examples to update the weights $\theta_t$, which computes the second gradient. In addition, the process of collecting adversarial example $\hat{x}_i^*$ and use $\hat{x}_i^*$ to update the model are tightly coupled, which means that each worker cannot calculate local loss $\mathbb{E}_{(x_i, y_i) \sim \mathbb{D}^k} \mathcal{L}(\theta_t; x_i, y_i)$ and $\mathbb{E}_{(x_i, y_i) \sim \mathbb{D}^k} \mathcal{L}(\theta_t; \hat{x}_i^*, y_i)$ to update the weights $\theta_t$, until the total adversarial examples $\hat{x}_i^*$ are obtained.

### 3.3 CONCURRENT ADVERSARIAL LEARNING FOR LARGE-BATCH TRAINING

As mentioned in the previous section, the vanilla DisAdv requires two sequential gradient computations at each step, where the first gradient computation is to obtain $\hat{x}_i^*$ based on $\mathcal{L}(\theta_t, x_i, y_i)$ and then compute the gradient of $\mathcal{L}(\theta_t, \hat{x}_i^*, y_i)$ to update $\theta_t$. Due to the sequential update nature, this overhead cannot be reduced even when increasing the number of processors — even with an infinite number of processors, the speed of two sequential computations will be twice of one parallel update. This makes adversarial learning unsuitable for large-batch training. In the following, we propose a simple but novel method to resolve this issue, and provide theoretical analysis on the proposed method.

**Concurrent Adversarial Learning (ConAdv)** As shown in Figure 3, our main finding is that if we use stale weights $(\theta_{t-\tau})$ for generating adversarial examples, then two sequential computations can

be de-coupled and the parameter update step run **concurrently** with the future adversarial example generation step.

Now we formally define the ConAdv procedure. Assume $x_i$ is sampled at iteration $t$, instead of the current weights $\theta_t$, we use stale weights $\theta_{t-\tau}$ (where $\tau$ is the delay) to calculate the gradient and further obtain an approximate adversarial example $\hat{x}_i(\theta_{t-\tau})$:

$$g_a(\theta_{t-\tau}) = \nabla_x \mathcal{L}(\theta_{t-\tau}; x_i, y_i), \quad \hat{x}_i(\theta_{t-\tau}) = x_i + \alpha \cdot g_a(\theta_{t-\tau}). \tag{3}$$

In this way, we can obtain the adversarial sample $\hat{x}_i(\theta_{t-\tau})$ through stale weights before updating the model at each step $t$. Therefore, the training efficiency can be improved. The structure of ConAdv is shown in Figure 1: At each step $t$, each worker $k$ can directly concatenate the clean mini-batch data and adversarial mini-batch data to calculate the gradient $\hat{g}_t^k(\theta_t)$ and update the model. That is because the system has obtained the approximate adversarial example $\hat{x}_i$ based on the stale weights $\theta_{t-\tau}$ before iteration $t$.

In practice, we set $\tau = 1$ so the adversarial examples $\hat{x}_i$ is computed at iteration $t - 1$. Therefore, each iteration will compute the current weight update and the adversarial examples for the next batch:

$$\theta_{t+1} = \theta_t + \frac{\eta}{2} \nabla_\theta (\mathbb{E}_{(x_i, y_i) \sim \mathcal{B}_{t,c}} \mathcal{L}(\theta_t; x_i, y_i) + \mathbb{E}_{\hat{x}_i, y_i \sim \mathcal{B}_{t,a}} \mathcal{L}(\theta_t, \hat{x}_i(\theta_{t-1}), y_i)), \tag{4}$$

$$\hat{x}_i(\theta_t) = x_i + \alpha \cdot \nabla_x \mathcal{L}(\theta_t; x_i, y_i), \quad \text{where} \quad (x_i, y_i) \in \mathcal{B}_{c,t+1}, \tag{5}$$

where $\mathcal{B}_{c,t} = \cup_{k=1}^{k=K} \mathcal{B}_{c,t}^k$ denotes clean mini-batch of all workers and $\mathcal{B}_{a,t} = \cup_{k=1}^{k=K} \mathcal{B}_{a,t}^k$ represents adversarial mini-batch of all workers. These two computations can be parallelized so there is no longer two sequential computations at each step. In large-batch setting when the number of workers reaches the limit that each batch size can use, ConAdv is similarly fast as standard optimizers such as SGD or Adam. The pseudo code of proposed ConAdv is shown in Algorithm 1.

## 3.4 CONVERGENCE ANALYSIS

In this section, we will show that despite using the stale gradients, ConAdv still enjoys nice convergence properties. For simplicity, we will use $\mathcal{L}(\theta, x_i)$ as a shorthand for $\mathcal{L}(\theta; x_i, y_i)$ and $\| \cdot \|$ indicates the $\ell_2$ norm. We define the optimal adversarial example as $x_i^* = \arg\max_{x_i^* \in \mathcal{X}_i} \mathcal{L}(\theta_t, x_i^*)$. In order to present our main theorem, we will need the following assumptions.

**Assumption 1.** *The function $\mathcal{L}(\theta, x)$ satisfies the Lipschitzian conditions:*

$$\|\nabla_x \mathcal{L}(\theta_1; x) - \nabla_x \mathcal{L}(\theta_2; x)\| \le L_{x\theta} \|\theta_1 - \theta_2\|, \|\nabla_\theta \mathcal{L}(\theta_1; x) - \nabla_\theta \mathcal{L}(\theta_2; x)\| \le L_{\theta\theta} \|\theta_1 - \theta_2\|,$$
$$\|\nabla_\theta \mathcal{L}(\theta; x_1) - \nabla_\theta \mathcal{L}(\theta; x_2)\| \le L_{\theta x} \|x_1 - x_2\|, \|\nabla_x \mathcal{L}(\theta; x_1) - \nabla_x \mathcal{L}(\theta; x_2)\| \le L_{xx} \|x_1 - x_2\|. \tag{6}$$

**Assumption 2.** $\mathcal{L}(\theta, x)$ *is locally $\mu$-strongly concave in* $\mathcal{X}_i = \{x^* : ||x^* - x_i||_\infty \le \epsilon\}$ *for all* $i \in [n]$, *i.e., for any* $x_1, x_2 \in \mathcal{X}_i$,

$$\mathcal{L}(\theta, x_1) \le \mathcal{L}(\theta, x_2) + \langle \nabla_x \mathcal{L}(\theta, x_2), x_1 - x_2 \rangle - \frac{\mu}{2} \|x_1 - x_2\|. \tag{7}$$

Assumption 2 can be verified based on the relationship between robust optimization and distributional robust optimization in (Sinha et al., 2017; Lee & Raginsky, 2017).

**Assumption 3.** *The concurrent stochastic gradient $\hat{g}(\theta_t) = \frac{1}{2|\mathcal{B}|} \sum_{i=1}^{|\mathcal{B}|} (\nabla_\theta \mathcal{L}(\theta_t; x_i) + \nabla_\theta \mathcal{L}(\theta_t, \hat{x}_i))$ is bounded by the constant $M$:*

$$\|\hat{g}(\theta_t)\| \le M. \tag{8}$$

**Assumption 4.** *Suppose* $\mathcal{L}_D(\theta_t) = \frac{1}{2n} \sum_{i=1}^{n} (\mathcal{L}(\theta_t, x_i^*) + \mathcal{L}(\theta_t, x_i))$, $g(\theta_t) = \frac{1}{2|\mathcal{B}|} \sum_{i=1}^{|\mathcal{B}|} (\nabla_\theta \mathcal{L}(x_i) + \nabla_\theta \mathcal{L}(\theta_t, x_i^*))$ *and* $\mathbb{E}[g(\theta_t)] = \nabla \mathcal{L}_D(\theta_t)$, *where* $|\mathcal{B}|$ *represents batch size . The variance of $g(\theta_t)$ is bounded by $\sigma^2$:*

$$\mathbb{E}[\|g(\theta_t) - \nabla \mathcal{L}_D(\theta_t)\|^2] \le \sigma^2. \tag{9}$$

Based on the above assumptions, we can obtain the upper bound between original adversarial example $x_i^*(\theta_t)$ and concurrent adversarial example $x_i^*(\theta_{t-\tau})$, where $\tau$ is the delay time.

**Lemma 1.** *Under Assumptions 1 and 2, we have*

$$\|x_i^*(\theta_t) - x_i^*(\theta_{t-\tau})\| \leq \frac{L}{\mu}\|\theta_t - \theta_{t-\tau}\| \leq \frac{L_{x\theta}}{\mu}\eta\tau M. \tag{10}$$

Lemma 1 illustrates the relation between $x_i^*(\theta_t)$ and $x_i^*(\theta_{t-\tau})$, which is bounded by the delay $\tau$. When the delay is small enough, $x_i^*(\theta_{t-\tau})$ can be regarded as an approximator of $x_i^*(\theta_t)$. We now establish the convergence rate as the following theorem.

**Theorem 1.** *Suppose Assumptions 1, 2, 3 and 4 hold. Let loss function $\mathcal{L}_D(\theta_t) = \frac{1}{2n}\sum_{i=1}^{n}(\mathcal{L}(\theta_t; x_i^*, y_i) + \mathcal{L}(\theta_t; x_i, y_i))$ and $\hat{x}_i(\theta_{t-\tau})$ be the $\lambda$-solution of $x_i^*(\theta_{t-\tau})$: $\langle x_i^*(\theta_{t-\tau}) - \hat{x}_i(\theta_{t-\tau}), \nabla_x\mathcal{L}(\theta_{t-\tau}; \hat{x}_i(\theta_{t-\tau}), y_i)\rangle \leq \lambda$. Under Assumptions 1 and 2, for the concurrent stochastic gradient $\hat{g}(\theta)$. If the step size of outer minimization is set to $\eta_t = \eta = \min(1/L, \sqrt{\Delta/L\sigma^2 T})$. Then the output of Algorithm 1 satisfies:*

$$\frac{1}{T}\sum_{t=0}^{T-1}\mathbb{E}[\|\nabla\mathcal{L}_D(\theta_t)\|_2^2] \leq 2\sigma\sqrt{\frac{L\Delta}{T}} + \frac{L_{\theta x}^2}{2}(\frac{\tau ML_{x\theta}}{L\mu} + \sqrt{\frac{\lambda}{\mu}})^2, \tag{11}$$

*where $L = L_{\theta\theta} + \frac{L_{x\theta}}{2\mu}L_{\theta x}$*

Our result provides a formal convergence rate of ConAdv and it can converge to a first-order stationary point at a sublinear rate up to a precision of $\frac{L_{\theta x}^2}{2}(\frac{\tau ML_{x\theta}}{L\mu} + \sqrt{\frac{\lambda}{\mu}})^2$, which is related to $\tau$. In practice we use the smallest delay $\tau = 1$ as discussed in the previous subsection.

## 4 EXPERIMENTAL RESULTS

### 4.1 EXPERIMENTAL SETUP

**Architectures and Datasets.** We select ResNet as our default architectures. More specially, we use the mid-weight version (ResNet-50) to evaluate the performance of our proposed algorithm. The dataset we used in this paper is ImageNet-1k, which consists of 1.28 million images for training and 50k images for testing. The convergence baseline of ResNet-50 in MLPerf is 75.9% top-1 accuracy in 90 epochs (i.e. ResNet-50 version 1.5 (Goyal et al., 2017)).

**Implementation Details.** We use TPU-v3 for all our experiments and the same setting as the baseline. We consider 90-epoch training for ResNet-50. For data augmentation, we mainly consider AutoAug (AA). In addition, we use LARS (You et al., 2017) to train all the models. Finally, for adversarial training, we always use 1-step PGD attack with random initialization.

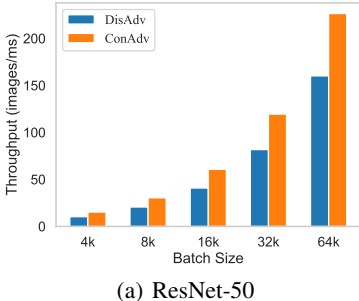
(a) ResNet-50

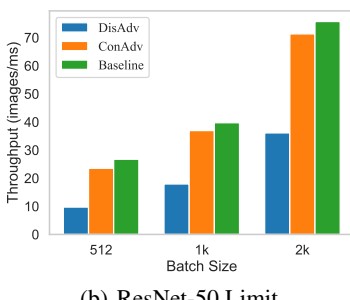
(b) ResNet-50 Limit

Figure 4: (a): throughput on scaling up batch size for ResNet-50, (b): throughtput when the number of processors reach the limit that each batch size can use for ResNet-50 .

### 4.2 IMAGENET TRAINING WITH RESNET

We train ResNet-50 with ConAdv and compare it with vanilla training and DisAdv. The experimental results of scaling up batch size in Table 1 illustrates that ConAdv can obtain the similar

accuracy compared with DisAdv and meanwhile speed up the training process. More specially, we can find that the top-1 accuracy of all methods are stable when the batch size is increased from 4k to 32k. After that, the performance starts to drop, which illustrates the bottleneck of large-batch training. However, ConAdv can improve the top-1 accuracy and the improved performance is stable as DisAdv does when the batch size reaches the bottleneck (such as 32k, 64k, 96k), but AutoAug gradually reaches its limitations. For instance, the top-1 accuracy increases from 74.3 to 75.3 when using ConAdv with a batch size of 96k and improved accuracy is 0.7%, 0.9% and 1.0% for 32k, 64k and 96k. However, AutoAug cannot further improve the top-1 accuracy when the batch size is 96k. The above results illustrate that adversarial learning can successfully maintain a good test performance in the large-batch training setting and can outperform data augmentation.

Table 1: Top-1 accuracy for ResNet-50 on ImageNet

| Method | 1k | 4k | 8k | 16k | 32k | 64k | 96k |
|---|---|---|---|---|---|---|---|
| **ResNet-50** | 76.9 | 76.9 | 76.6 | 76.6 | 76.6 | 75.3 | 74.3 |
| **ResNet-50+AA** | **77.5** | **77.5** | **77.4** | 77.1 | 76.9 | 75.6 | 74.3 |
| **ResNet-50+DisAdv** | 77.4 | 77.4 | **77.4** | **77.4** | **77.3** | **76.2** | **75.3** |
| **ResNet-50+ConAdv** | 77.4 | 77.4 | **77.4** | **77.4** | **77.3** | **76.2** | **75.3** |

In addition, Figure 4(a) presents the throughput (images/ms) on scaling up batch size. We can observe that ConAdv can further increase throughput and accelerate the training process. To obtain accurate statistics of BatchNorm, we need to make sure each worker has at least 64 examples to calculate them (Normal Setting). Thus, the number of cores is [Batch Size / 64]. For example, we use TPU v3-256 to train DisAdv when batch size is 32k, which has 512 cores (32k/64=512). As shown in Figure 4(a), the throughput of DisAdv increases from 10.3 on 4k to 81.7 on 32k and CondAdv achieve about 1.5x speedup compared with DisAdv, which verifies our proposed ConAdv can maintain the accuracy of large-batch training and meanwhile accelerate the training process.

To simulate the speedup when the number of workers reach the limit that each Batch Size can use, we use a large enough distributed system to train the model with the batch size of 512, 1k and 2k on TPU v3-128, TPU v3-256 and TPU v3-512 , respectively. The result is shown in Figure 4(b), we can obtain that ConAdv can achieve about 2x speedup compared with DisAdv. Furthermore, in this scenario we can observe ConAdv can achieve the similar throughput as Baseline (vanilla ResNet-50 training). For example, compared with DisAdv, the throughput increases from 36.1 to 71.3 when using ConAdv with a batch size of 2k. In addition, the throughput is 75.7, which illustrates that ConAdv can achieve a similar speed as baseline. However, ConAdv can expand to larger batch size than baseline. Therefore, ConAdv can further accelerate the training of deep neural network.

### 4.3 IMAGENET TRAINING WITH DATA AUGMENTATION

To explore the limit of our method and evaluate whether adversarial learning can be combined with data augmentation for large-batch training, we further apply data augmentation into the proposed adversarial learning algorithm and the results are shown in Table 2. We can find that ConAdv can further improve the performance of large-batch training on ImageNet when combined with Autoaug (AA). Under this setting, we can expand the batch size to more than 96k, which can improve the algorithm efficiency and meanwhile benefit the machine utilization. For instance, for ResNet, the top-1 accuracy increases from 74.3 to 76.2 under 96k when using ConAdv and AutoAug.

Table 2: Top-1 accuracy with AutoAug on ImageNet

| Method | 1k | 4k | 8k | 16k | 32k | 64k | 96k |
|---|---|---|---|---|---|---|---|
| **ResNet-50** | 76.9 | 76.9 | 76.6 | 76.6 | 76.6 | 75.3 | 74.3 |
| **ResNet-50+AA** | 77.5 | 77.5 | 77.4 | 77.1 | 76.9 | 75.6 | 74.3 |
| **ResNet-50+ConAdv+AA** | **78.5** | **78.5** | **78.5** | **78.5** | **78.3** | **77.3** | **76.2** |

### 4.4 TRAINING TIME

The wall clock training times for ResNet-50 are shown in Table 3. We can find that the training time gradually decreases with batch size increasing. For example, the training time of ConAdv decreases

from 1277s to 227s when scale the batch size from 16k to 64k. In addition, we can find that DisAdv need about 1.5x training time compared with vanilla ResNet-50 but ConAdv can efficiently reduce the training of DisAdv to a level similar to vanilla ResNet. For instance, the training time of DisAdv is reduced from 1191s to 677s when using ConAdv. Noted that we don't

Table 3: Training Time Analysis

| Method | 16k | 32k | 64k |
|---|---|---|---|
| **ResNet-50** | **1194s** | **622s** | / |
| **DisAdv** | 1657s | 1191s | 592s |
| **ConAdv** | 1277s | 677s | **227s** |

report the clock time for vanilla ResNet-50 at 64k since the top-1 accuracy is below the MLPerf standard 75.9%. The number of machines required to measure the maximum speed at 96K exceeds our current resources. The comparison on 32k and 64k also can evaluate the runtime improvement.

## 4.5 GENERALIZATION GAP

To the best of our knowledge, theoretical analysis of generalization errors in large-batch setting is still an open problem. However, we empirically found that our method can successfully reduce the generalization gap in large-batch training. The experimental results in Table 4 indicate that ConAdv can narrow the generalization gap. For example, the generalization gap is 4.6 for vanilla ResNet-50 at 96k and ConAdv narrows the gap to 2.7. In addition, combining ConAdv with AutoAug, the training accuracy and test accuracy can further increase and meanwhile maintain the similar generalization gap.

Table 4: Generalization Gap of Large-Batch Training on ImageNet-1k

| | Vanilla ResNet-50 | | | | ConAdv | | | | ConAdv + AA | | | |
|---|---|---|---|---|---|---|---|---|---|---|---|---|
| | 16k | 32k | 64k | 96k | 16k | 32k | 64k | 96k | 16k | 32k | 64k | 96k |
| **Training Accuracy** | 81.4 | 82.5 | 79.6 | 78.9 | 80.3 | 80.8 | 78.2 | 78.0 | 81.6 | 81.7 | 79.6 | 78.4 |
| **Test Accuracy** | 76.6 | 76.6 | 75.3 | 74.3 | 77.4 | 77.3 | 76.2 | 75.3 | 78.5 | 78.3 | 77.3 | 76.2 |
| **Generalization Gap** | 4.8 | 5.9 | 4.3 | 4.6 | 2.9 | 3.5 | 2.0 | 2.7 | 3.1 | 3.4 | 2.3 | 2.2 |

## 4.6 ANALYSIS OF ADVERSARIAL PERTURBATION

Adversarial learning calculates an adversarial perturbation on input data to smooth the decision boundary and help the model converge to the flat minima. In this section, we analyze the effects of different perturbation values for the performance of large-batch training on ImageNet. The analysis results are illustrated in Table 5. It presents that we should increase the attack intensity as the batch size increasing. For example, the best attack perturbation value increases from 3 (32k) to 7 (96k) for ResNet-50 and from 8 (16k) to 12 (64k). In addition, we should increase the perturbation value when using data augmentation. For example, the perturbation value should be 3 for the original ResNet-50 but be 5 when data augmentation is applied.

Table 5: Experiment Results (Top-1 Accuracy) when useing Different Adversarial Perturbation.

| Method | Batch Size | p=0 | p=1 | p=2 | p=3 | p=4 | p=5 | p=6 | p=7 | p=8 | p=9 | p=10 | p=12 |
|---|---|---|---|---|---|---|---|---|---|---|---|---|---|
| **ResNet-50 + ConAdv** | 32K | 76.8 | 77.2 | 77.3 | **77.4** | 77.3 | 77.3 | 77.3 | 77.3 | 77.3 | 77.3 | 77.2 | 77.2 |
| **ResNet-50 + ConAdv + AA** | 32K | 77.8 | 78.0 | 78.1 | 78.1 | 78.0 | **78.3** | 78.2 | 78.2 | 78.2 | 78.2 | 78.2 | 78.1 |
| **ResNet-50 + ConAdv** | 64K | 75.7 | 76.2 | 76.3 | 76.3 | 76.4 | **76.7** | 76.4 | 76.4 | 76.4 | 76.4 | 76.4 | 76.3 |
| **ResNet-50 + ConAdv + AA** | 64K | 76.8 | 77.0 | 76.8 | 77.0 | 77.1 | 77.1 | 77.2 | **77.4** | 77.2 | 77.1 | 77.1 | 77.1 |
| **ResNet-50 + ConAdv** | 96K | 74.6 | 75.1 | 75.1 | 75.1 | 75.3 | 75.1 | 75.1 | 75.1 | **75.3** | 75.2 | 75.1 | 75.1 |
| **ResNet-50 + ConAdv + AA** | 96K | 75.8 | 75.9 | 75.8 | 76.0 | 76.0 | 76.0 | 76.0 | 76.1 | **76.2** | **76.2** | 76.0 | 76.0 |

## 5 CONCLUSIONS

We firstly analyze the effect of data augmentation for large-batch training and propose a novel distributed adversarial learning algorithm to scale to a larger batch size. To reduce the overhead of adversarial learning, we further propose a novel concurrent adversarial learning to decouple the two sequential gradient computations in adversarial learning. We evaluate our proposed method on ResNet. The experimental results show that our proposed method is beneficial for large-batch training.

# 6 ACKNOWLEDGEMENTS

We thank Google TFRC for supporting us to get access to the Cloud TPUs. We thank CSCS (Swiss National Supercomputing Centre) for supporting us to get access to the Piz Daint supercomputer. We thank TACC (Texas Advanced Computing Center) for supporting us to get access to the Longhorn supercomputer and the Frontera supercomputer. We thank LuxProvide (Luxembourg national super-computer HPC organization) for supporting us to get access to the MeluXina supercomputer. CJH and XC are partially supported by NSF under IIS-2008173, IIS-2048280 and by Army Research Laboratory under agreement number W911NF-20-2-0158.

# 7 ETHICS STATEMENT

We do not have any potential ethics issues in this paper. We hope to propose a novel distributed adversarial learning algorithm to accelerate large-batch training.

# 8 REPRODUCIBILITY STATEMENT

we list our main hyperparameters for large-batch training in Appendix A.4 (Table 6). For experimental details, we introduce our experiment settings in section 4, such as the dataset, model architecture, data augmentation, optimizer and so on.

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

# A APPENDIX

## A.1 THE PROOF OF LEMMA 1:

This completes the proof. The proof is inspired by Sinha et al. (2017); Wang et al. (2019). Under Assumptions 1 and 2, we have

$$\|x_i^*(\theta_t) - x_i^*(\theta_{t-\tau})\| \leq \frac{L_{x\theta}}{\mu}\|\theta_t - \theta_{t-\tau}\| \leq \frac{L_{x\theta}}{\mu}\eta\tau M \tag{12}$$

where $x_i^*(\theta_t)$ and $x_i^*(\theta_{t-\tau})$ denote the adversarial example of $x_i$ calculated by $\theta_t$ and stale weight $\theta_{t-\tau}$, respectively.

Proof:

According to Assumption 2, we have

$$\begin{aligned}
\mathcal{L}(\theta_t, x_i^*(\theta_{t-\tau})) \leq{} & \mathcal{L}(\theta_t, x_i^*(\theta_t)) + \langle\nabla_x\mathcal{L}(\theta_t, x_i^*(\theta_t)), x_i^*(\theta_{t-\tau}) - x_i^*(\theta_t)\rangle - \\
& \frac{\mu}{2}\|x_i^*(\theta_{t-\tau}) - x_i^*(\theta_t)\|^2, \\
\leq{} & \mathcal{L}(\theta_t, x_i^*(\theta_t)) - \frac{\mu}{2}\|x_i^*(\theta_{t-\tau}) - x_i^*(\theta_t)\|^2
\end{aligned} \tag{13}$$

In addition, we have

$$\begin{aligned}
\mathcal{L}(\theta_t, x_i^*(\theta_t)) \leq{} & \mathcal{L}(\theta_t, x_i^*(\theta_{t-\tau})) + \langle\nabla_x\mathcal{L}(\theta_t, x_i^*(\theta_{t-\tau})), x_i^*(\theta_t) - x_i^*(\theta_{t-\tau})\rangle - \\
& \frac{\mu}{2}\|x_i^*(\theta_{t-\tau}) - x_i^*(\theta_t)\|^2
\end{aligned} \tag{14}$$

Combining (13) and (14), we can obtain:

$$\begin{aligned}
\mu\|x_i^*(\theta_{t-\tau}) - x_i^*(\theta_t)\|^2 &\leq \langle\nabla_x\mathcal{L}(\theta_t, x_i^*(\theta_t)), x_i^*(\theta_t) - x_i^*(\theta_{t-\tau})\rangle \\
&\leq \langle\nabla_x\mathcal{L}(\theta_t, x_i^*(\theta_{t-\tau})) - \nabla_x\mathcal{L}(\theta_{t-\tau}, x_i^*(\theta_{t-\tau})), x_i^*(\theta_t) - x_i^*(\theta_{t-\tau})\rangle \\
&\leq \|\nabla_x\mathcal{L}(\theta_t, x_i^*(\theta_{t-\tau})) - \nabla_x\mathcal{L}(\theta_{t-\tau}, x_i^*(\theta_{t-\tau}))\|\|x_i^*(\theta_t) - x_i^*(\theta_{t-\tau})\| \\
&\leq L_{x\theta}\|\theta_t - \theta_{t-\tau}\|\|x_i^*(\theta_t) - x_i^*(\theta_{t-\tau})\|
\end{aligned} \tag{15}$$

where the second inequality is due to $\langle\nabla_x\mathcal{L}(\theta_{t-\tau}, x_i^*(\theta_{t-\tau})), x_i^*(\theta_t) - x_i^*(\theta_{t-\tau})\rangle \leq 0$, the third inequality holds because CauchySchwarz inequality and the last inequality follows from Assumption 1. Therefore,

$$\begin{aligned}
\|x_i^*(\theta_t) - x_i^*(\theta_{t-\tau})\| &\leq \frac{L_{x\theta}}{\mu}\|\theta_t - \theta_{t-\tau}\| \\
&\leq \frac{L_{x\theta}}{\mu}\|\sum_{j\in[1,\tau]}(\theta_{t-j+1} - \theta_{t-j})\| \\
&\leq \frac{L_{x\theta}}{\mu}\|\sum_{j\in[1,\tau]}\eta\hat{g}_{t-j}(x_i))\| \\
&\leq \frac{L_{x\theta}}{\mu}\eta\tau M
\end{aligned} \tag{16}$$

where the second inequality follows the calculation of delayed weight, the third inequality holds because the difference of weights is calculated with gradient $\hat{g}_{t-j}(j\in[1,\tau])$ and the last inequality holds follows Assumption 3.

Thus,

$$\|x_i^*(\theta_t) - x_i^*(\theta_{t-\tau})\| \leq \frac{L_{x\theta}}{\mu}\eta\tau M \tag{17}$$

This completes the proof.

A.2

**Lemma 2.** *Under Assumptions 1 and 2, we have $\mathcal{L}_D(\theta)$ is L-smooth where $L = L_{\theta\theta} + \frac{L_{x\theta}}{2\mu}L_{\theta x}$, i.e., for any $\theta_1$ and $\theta_2$, we can say*

$$\|\nabla_\theta \mathcal{L}_D(\theta_t) - \nabla_\theta \mathcal{L}_D(\theta_{t-\tau})\| \leq L\|\theta_t - \theta_{t-\tau}\| \tag{18}$$

$$\mathcal{L}_D(\theta_t) = \mathcal{L}_D(\theta_{t-\tau}) + \langle \mathcal{L}_D(\theta_{t-\tau}), \theta_t - \theta_{t-\tau}\rangle + \frac{L}{t-\tau}\|\theta_t - \theta_{t-\tau}\| \tag{19}$$

Proof:

Based on Lemma 1, we can obtain:

$$\|x_i^*(\theta_t) - x_i^*(\theta_{t-\tau})\| \leq \frac{L_{x\theta}}{\mu}\|\theta_t - \theta_{t-\tau}\| \tag{20}$$

We can obtain for $i \in [n]$:

$$\begin{aligned}
\|\nabla_\theta \mathcal{L}(\theta_t, &x_i^*(\theta_t)) - \nabla_\theta \mathcal{L}(\theta_{t-\tau}, x_i^*(\theta_{t-\tau}))\|\\
\leq & \|\nabla_\theta \mathcal{L}(\theta_t, x_i^*(\theta_t)) - \nabla_\theta \mathcal{L}(\theta_t, x_i^*(\theta_{t-\tau}))\|\\
& + \|\nabla_\theta \mathcal{L}(\theta_t, x_i^*(\theta_{t-\tau})) - \nabla_\theta \mathcal{L}(\theta_{t-\tau}, x_i^*(\theta_{t-\tau}))\|\\
\leq & L_{\theta\theta}\|\theta_t - \theta_{t-\tau}\| + L_{\theta x}\|x_i^*(\theta_t) - x_i^*(\theta_{t-\tau})\|\\
\leq & L_{\theta\theta}\|\theta_t - \theta_{t-\tau}\| + L_{\theta x}\frac{L_{x\theta}}{\mu}\|\theta_t - \theta_{t-\tau}\|\\
= & (L_{\theta\theta} + L_{\theta x}\frac{L_{x\theta}}{\mu})\|\theta_t - \theta_{t-\tau}\|
\end{aligned} \tag{21}$$

where the second inequality holds because Assumption 1, the third inequality holds follows Lemma 1.

Therefore,

$$\begin{aligned}
\|\nabla\mathcal{L}_D(\theta_t) - \nabla\mathcal{L}_D(\theta_{t-\tau})\|_2 \leq & \|\frac{1}{2n}\sum_{i=1}^{n}(\nabla_\theta \mathcal{L}(\theta_t, x_i) + \nabla_\theta \mathcal{L}(\theta_t, x_i^*(\theta_t)))\\
& - \frac{1}{2n}\sum_{i=1}^{n}(\nabla_\theta \mathcal{L}(\theta_{t-\tau}, x_i) + \nabla_\theta \mathcal{L}(\theta_{t-\tau}, x_i^*(\theta_{t-\tau})))\|\\
\leq & \frac{1}{2n}\sum_{i=1}^{n}\|\nabla_\theta \mathcal{L}(\theta_t, x_i) - \nabla_\theta \mathcal{L}(\theta_{t-\tau}, x_i)\|\\
& + \frac{1}{2n}\sum_{i=1}^{n}\|\nabla_\theta \mathcal{L}(\theta_t, x_i^*(\theta_t)) - \nabla_\theta \mathcal{L}(\theta_{t-\tau}, x_i^*(\theta_{t-\tau}))\|\\
\leq & \frac{1}{2}L_{\theta\theta}\|\theta_t - \theta_{t-\tau}\| + \frac{1}{2}(L_{\theta\theta} + L_{\theta x}\frac{L_{x\theta}}{\mu})\|\theta_t - \theta_{t-\tau}\|\\
= & (L_{\theta\theta} + \frac{L_{x\theta}}{2\mu}L_{\theta x})\|\theta_t - \theta_{t-\tau}\|
\end{aligned} \tag{22}$$

This completes the proof.

A.3

**Lemma 3.** *Let $\hat{x}_i(\theta_t)$ be the $\lambda$-solution of $x_i^*(\theta_t)$: $\langle x_i^*(\theta_t) - \hat{x}_i(\theta_t), \nabla_x \mathcal{L}(\theta_t, \hat{x}_i(\theta_t)) \rangle \leq \lambda$. Under Assumptions 1 and 2, for the concurrent stochastic gradient $\hat{g}(\theta)$, we have*

$$\|g(\theta_t) - \hat{g}(\theta_{t-\tau})\| \leq \frac{L_{\theta x}}{2} \sqrt{\frac{\lambda}{\mu}}. \tag{23}$$

Proof:

$$
\begin{aligned}
\|g(\theta_t) - \hat{g}(\theta_{t-\tau})\| &= \|\frac{1}{2|\mathcal{B}|} \sum_{i \in |\mathcal{B}|} (\nabla_\theta \mathcal{L}(\theta_t, x_i) + \nabla_\theta \mathcal{L}(\theta_t, x_i^*(\theta_t))) \\
&\quad - (\nabla_\theta \mathcal{L}(\theta_t, x_i) + \nabla_\theta \mathcal{L}(\theta_t, \hat{x}_i(\theta_{t-\tau})))\| \\
&= \|\frac{1}{2|\mathcal{B}|} \sum_{i \in |\mathcal{B}|} (\nabla_\theta \mathcal{L}(\theta_t, x_i^*(\theta_t)) - \nabla_\theta \mathcal{L}(\theta_t, \hat{x}_i(\theta_{t-\tau})))\| \\
&\leq \frac{1}{2|\mathcal{B}|} \sum_{i \in |\mathcal{B}|} \|\nabla_\theta \mathcal{L}(\theta_t, x_i^*(\theta_t)) - \nabla_\theta \mathcal{L}(\theta_t, \hat{x}_i(\theta_{t-\tau}))\| \\
&\leq \frac{1}{2|\mathcal{B}|} \sum_{i \in |\mathcal{B}|} L_{\theta x} \|x_i^*(\theta_t) - \hat{x}_i(\theta_{t-\tau})\| \\
&= \frac{1}{2|\mathcal{B}|} \sum_{i \in |\mathcal{B}|} L_{\theta x} \|x_i^*(\theta_t) - x_i^*(\theta_{t-\tau}) + x_i^*(\theta_{t-\tau}) - \hat{x}_i(\theta_{t-\tau})\| \\
&\leq \frac{1}{2|\mathcal{B}|} \sum_{i \in |\mathcal{B}|} (L_{\theta x} \|x_i^*(\theta_t) - x_i^*(\theta_{t-\tau})\| + L_{\theta x} \|x_i^*(\theta_{t-\tau}) - \hat{x}_i(\theta_{t-\tau})\|) \\
&= \frac{1}{2|\mathcal{B}|} \sum_{i \in |\mathcal{B}|} (L_{\theta x} \|x_i^*(\theta_t) - x_i^*(\theta_{t-\tau})\| + L_{\theta x} \|x_i^*(\theta_{t-\tau}) - \hat{x}_i(\theta_{t-\tau})\|) \\
&\leq \frac{1}{2|\mathcal{B}|} \sum_{i \in |\mathcal{B}|} (L_{\theta x} \eta \tau M \frac{L_{x\theta}}{\mu} + L_{\theta x} \|x_i^*(\theta_{t-\tau}) - \hat{x}_i(\theta_{t-\tau})\|)
\end{aligned}
\tag{24}
$$

Let $\hat{x}_i(\theta_{t-\tau})$ be the $\lambda$-approximate of $x_i^*(\theta_{t-\tau})$, we can obtain:

$$\langle x_i^*(\theta_{t-\tau}) - \hat{x}_i(\theta_{t-\tau}), \nabla_\theta \mathcal{L}(\theta_{t-\tau}; \hat{x}_i(\theta_{t-\tau})) \rangle \leq \delta \tag{25}$$

In addition, we can obtain:

$$\langle \hat{x}_i(\theta_{t-\tau}) - x_i^*(\theta_{t-\tau}), \nabla_x \mathcal{L}(\theta_{t-\tau}, x_i^*(\theta_{t-\tau})) \rangle \leq 0 \tag{26}$$

Combining 25 and 26, we have:

$$\langle x_i^*(\theta_{t-\tau}) - \hat{x}_i(\theta_{t-\tau}), \nabla_\theta \mathcal{L}(\theta_{t-\tau}; \hat{x}_i(\theta_{t-\tau})) - \nabla_x \mathcal{L}(\theta_{t-\tau}, x_i^*(\theta_{t-\tau})) \rangle \leq \lambda \tag{27}$$

Based on Assumption 2, we have

$$\mu \|x_i^*(\theta_{t-\tau}) - \hat{x}_i(\theta_{t-\tau})\|^2 \leq \langle \nabla_x \mathcal{L}(\theta_{t-\tau}, x_i^*(\theta_{t-\tau})) - \nabla_x \mathcal{L}(\theta_{t-\tau}, \hat{x}_i(\theta_{t-\tau}), \hat{x}_i - x_i^*(\theta_{t-\tau})) \rangle \tag{28}$$

Combining 28 with 27, we can obtain:

$$\mu \|x_i^*(\theta_{t-\tau}) - \hat{x}(\theta_{t-\tau})\|^2 \leq \lambda \tag{29}$$

Therefore, we have

$$\|x_i^*(\theta_{t-\tau}) - \hat{x}(\theta_{t-\tau})\| \leq \sqrt{\frac{\lambda}{\mu}}. \tag{30}$$

Thus, we can obtain

$$\|g(\theta_t) - \hat{g}(\theta_{t-\tau})\| \leq \frac{L_{\theta x}}{2}(\eta \tau M \frac{L_{x\theta}}{\mu} + \sqrt{\frac{\lambda}{\mu}}) \tag{31}$$

This completes the proof.

### A.3 THE PROOF OF THEOREM 1:

Suppose Assumptions 1, 2, 3 and 4 hold. Let $\Delta = \mathcal{L}_D(\theta_0) - \min_\theta \mathcal{L}_D(\theta)$, $\nabla \mathcal{L}_D(\theta_t) = \frac{1}{2n}\sum_{i=1}^n (\nabla \mathcal{L}(\theta_t, x_i, y_i) + \nabla \mathcal{L}(\theta_t, x_i^*, y_i))$. If the step size of outer minimization is set to $\eta_t = \eta = \min(1/L, \sqrt{\Delta/L\sigma^2 T})$, where $L = L_{\theta\theta} + \frac{L_{x\theta}}{2\mu}L_{\theta x}$. Then the output of Algorithm 1 satisfies:

$$\frac{1}{T}\sum_{t=0}^{T-1}\mathbb{E}[\|\nabla \mathcal{L}_D(\theta_t)\|^2] \leq 2\sigma\sqrt{\frac{L\Delta}{T}} + \frac{L_{\theta x}^2}{2}(\frac{\tau M L_{x\theta}}{L\mu} + \sqrt{\frac{\lambda}{\mu}})^2 \tag{32}$$

where $L = (M L_{\theta x} L_{x\theta}/\epsilon\mu + L_{\theta\theta})$.

Proof:

$$
\begin{aligned}
\mathcal{L}_D(\theta_{t+1}) &\leq \mathcal{L}_D(\theta_t) + \langle \nabla \mathcal{L}_D(\theta_t), \theta_{t+1} - \theta_t \rangle + \frac{L}{2}\|\theta_{t+1} - \theta_t\|^2 \\
&= \mathcal{L}_D(\theta_t) - \eta\|\nabla \mathcal{L}_D(\theta_t)\|^2 + \frac{L\eta^2}{2}\|\hat{g}(\theta_t)\|_2^2 + \eta\langle \nabla \mathcal{L}_D(\theta_t), \nabla \mathcal{L}_D(\theta_t) - \hat{g}(\theta_t)\rangle \\
&= \mathcal{L}_D(\theta_t) - \eta(1 - \frac{L\eta}{2})\|\nabla \mathcal{L}_D(\theta_t)\|^2 + \eta(1 - L\eta)\langle \nabla \mathcal{L}_D(\theta_t), \nabla \mathcal{L}_D(\theta_t) - \hat{g}(\theta_t)\rangle \\
&\quad + \frac{L\eta^2}{2}\|\hat{g}(\theta_t) - \nabla \mathcal{L}_D(\theta_t)\|^2 \\
&= \mathcal{L}_D(\theta_t) - \eta(1 - \frac{L\eta}{2})\|\nabla \mathcal{L}_D(\theta_t)\|^2 + \eta(1 - L\eta)\langle \nabla \mathcal{L}_D(\theta_t), g(\theta_t) - \hat{g}(\theta_t)\rangle \\
&\quad + \eta(1 - L\eta)\langle \nabla \mathcal{L}_D(\theta_t), \nabla \mathcal{L}_D(\theta_t) - g(\theta_t)\rangle + \frac{L\eta^2}{2}\|\hat{g}(\theta_t) - g(\theta_t) + g(\theta_t) - \nabla \mathcal{L}_D(\theta_t)\|^2 \\
&\leq \mathcal{L}_D(\theta_t) - \frac{\eta}{2}\|\nabla \mathcal{L}_D(\theta_t)\|_2^2 + \frac{\eta}{2}(1 - L\eta)\|\hat{g}(\theta_t) - g(\theta_t)\|^2 \\
&\quad + \eta(1 - L\eta)\langle \nabla \mathcal{L}_D(\theta_t), \nabla \mathcal{L}_D(\theta_t) - g(\theta_t)\rangle + L\eta^2(\|\hat{g}(\theta_t) - g(\theta)\|_2^2 + \|g(\theta_t) - \nabla \mathcal{L}_D(\theta_t)\|^2) \\
&= \mathcal{L}_D(\theta_t) - \frac{\eta}{2}\|\nabla \mathcal{L}_D(\theta_t)\|_2^2 + \frac{\eta}{2}(1 + L\eta)\|\hat{g}(\theta_t) - g(\theta_t)\|^2 \\
&\quad + \eta(1 - L\eta)\langle \nabla \mathcal{L}_D(\theta_t), \nabla \mathcal{L}_D(\theta_t) - g(\theta_t)\rangle + L\eta^2\|g(\theta_t) - \nabla \mathcal{L}_D(\theta_t)\|_2^2)
\end{aligned}
\tag{33}
$$

Taking expectation on both sides of the above inequality conditioned on $\theta_t$, we can obtain:

$$\mathbb{E}[\mathcal{L}_D(\theta_{t+1}) - \mathcal{L}_D(\theta_t)|\theta_t] \leq -\frac{\eta}{2}\|\nabla\mathcal{L}_D(\theta_t)\|^2 + \frac{\eta}{2}(1+L\eta)(\frac{L_{\theta x}}{2}(\eta\tau M\frac{L_{x\theta}}{\mu} + \sqrt{\frac{\lambda}{\mu}}))^2 + L\eta^2\sigma^2$$

$$= -\frac{\eta}{2}\|\nabla\mathcal{L}_D(\theta_t)\|^2 + \frac{\eta}{8}(1+L\eta)(L_{\theta x}(\eta\tau M\frac{L_{x\theta}}{\mu} + \sqrt{\frac{\lambda}{\mu}}))^2 + L\eta^2\sigma^2$$

$$= -\frac{\eta}{2}\|\nabla\mathcal{L}_D(\theta_t)\|^2 + \frac{\eta L_{\theta x}^2}{8}(1+L\eta)(\eta\tau M\frac{L_{x\theta}}{\mu} + \sqrt{\frac{\lambda}{\mu}})^2 + L\eta^2\sigma^2$$

$$(34)$$

where we used the fact that $\mathbb{E}[g(\theta_t)] = \nabla\mathcal{L}_D(\theta_t)$, Assumption 2, Lemma 2 and Lemma 3. Taking the sum of (34) over $t = 0, ..., T-1$, we obtain that:

$$\sum_{t=0}^{T-1}\frac{\eta}{2}\mathbb{E}[\|\nabla\mathcal{L}_D(\theta_t)\|^2] \leq \mathbb{E}[\mathcal{L}_D(\theta_0) - \mathcal{L}_D(\theta_T)] + \sum_{t=0}^{T-1}\frac{\eta L_{\theta x}^2}{8}(1+L\eta)(\eta\tau M\frac{L_{x\theta}}{\mu} + \sqrt{\frac{\lambda}{\mu}})^2 + L\sum_{t=0}^{T-1}\eta^2\sigma^2$$

$$(35)$$

Choose $\eta = \min(1/L, \sqrt{\frac{\Delta}{TL\sigma^2}})$ where $\Delta = \mathcal{L}_D(\theta_0) - \mathcal{L}_D(\theta_T)$ and $L = L_{\theta\theta} + \frac{L_{x\theta}}{2\mu}L_{\theta x}$, we can show that:

$$\frac{1}{T}\sum_{t=0}^{T-1}\mathbb{E}[\|\nabla\mathcal{L}_D(\theta_t)\|^2] \leq 2\sigma\sqrt{\frac{L\Delta}{T}} + \frac{L_{\theta x}^2}{2}(\frac{\tau M L_{x\theta}}{L\mu} + \sqrt{\frac{\lambda}{\mu}})^2 \qquad (36)$$

## A.4 HYPERPARAMETERS

HYPERPARAMETERS:

More specially, our main hyperparameters are shown in Table 6.

Table 6: Hyperparameters of ResNet-50 on ImageNet

|  | 32k | 64k | 96k |
|---|---|---|---|
| **Peak LR** | 35.0 | 41.0 | 43.0 |
| **Epoch** | 90 | 90 | 90 |
| **Weight Decay** | 5E-4 | 5E-4 | 5E-4 |
| **Warmup** | 40 | 41 | 41 |
| **LR decay** | POLY | POLY | POLY |
| **Optimizer** | LARS | LARS | LARS |
| **Momentum** | 0.9 | 0.9 | 0.9 |
| **Label Smoothing** | 0.1 | 0.1 | 0.1 |

