# OpenReview forum: "Concurrent Adversarial Learning for Large-Batch Training"
_ICLR.cc/2022/Conference — ICLR 2022 Poster_

### Official Review · Reviewer_M9ZZ · 2021-10-30

**Correctness:** 3
**Technical Novelty And Significance:** 3
**Empirical Novelty And Significance:** 3
**Recommendation:** 6
**Confidence:** 5

**Main Review:**

The paper is clearly motivated and well written. To the best of my knowledge, the findings regarding the adversarial training for improving generalization performance in **large-batch training** is new. Although the proposed method is simple and straightforward, and the theoretical analysis is easy to conduct, the empirical results are nice.

The authors conducted various experiments, in which many empirical findings are quite interesting. However, the results lack of reasonable interpretations. For example,
1.	About the perturbation magnitude. From the experiment result in Table 5, it shows we should increase the attack intensity as the batch size increasing and augmentation used. Could you elaborate a little bit why should be this? It is a well known conclusion that there exists accuracy-robustness trade-off when using adversarial training, which means that larger perturbation might ruin the clean accuracy. This appears contrary to your finding. It is a really interesting point.
2.	Another important side regarding the training details is that whether the pure adversarial training helps, i.e. not mixing the clean data and adversarial data for training the networks. I think the authors should provide this result to deepen the understanding the role of adversarial examples for generalization.


**Summary Of The Paper:**

This manuscript empirically show that adversarial training in large-batch training scenario has better performance than traditional data augmentation. And furthermore, the authors proposed a simple method to conduct adversarial example generation and gradient computation w.r.t. to weights concurrently to accelerate the adversarial training in distributed setting. The key strategy is to use staled weights to generate adversarial examples, and then decoupled the bi-level optimization.

**Summary Of The Review:**

New empirical findings regarding the advesarial training helps generalization in large-batch training scenarios, but some results lack of deeper understanding.

---

> ### Author Response · Authors · 2021-11-21
> **Response to Reviewer M9ZZ**
>
> Thanks for your constructive comments.
>
> **[Increase attack intensity]** This phenomenon is possible because the model is easier to converge to sharp local minima when scaling up the batch size. Therefore, we need to increase attack intensity to improve generalization.
>
> We agree with you that adversarial learning usually maintains a trade-off between accuracy-robustness. However, our experimental results are not contrary to the above conclusion. Firstly,
> as for the larger perturbation might ruin the clean accuracy, these experiments usually fix the value of batch size. We find that we need to increase the perturbation magnitude when scaling up batch size in this paper. In addition, for large-batch training,
> we still observe the trade-off between accuracy and robustness. Specifically, there exists a best perturbation value for collecting adversarial examples. When the perturbation is larger than the threshold, the accuracy begins to drop.
>
>
> As shown in the Table 5 of the paper, the best perturbation value is p=3 at 32k and the corresponding accuracy is 77.4. However, when we further increase the attack intensity, the accuracy drop to 77.3 (p=4) and 77.2 (p=10). And the accuracy will further decrease when we keep increasing p.
> We think this phenomenon can explain the trade-off between accuracy and robustness.
>
> **[Pure adversarial training]**  We try to conduct the experiments on pure adversarial learning and the experimental results are shown in the table:
>
>  |Model     | 16k        | 32k           | 64k  |
> |:-------------| :-------------: |:-------------:| :----------------:|
> | **ResNet-50**      |  76.6 | 76.6 | 75.3   |
> | **ResNet-50+PureAdv**      | 55.3  | 55.9  | 47.9  |
> | **ResNet-50+ConAdv** | 77.4  | 77.3 | 76.2  |
>
> From this table, we can find that pure adversarial training will hurt the test accuracy. We think the reason is that the distribution gap between adversarial data and test data is large, which will affect the calculation of statistics in BatchNorm. More specially, pure adversarial learning only uses adversarial data to train the model. In this way, the moving mean and variance in BatchNorm is calculated only by adversarial data. However, there exists a distribution gap between clean data and adversarial data [1], which causes the accuracy drop.
>
> To verify the above analysis, we try to use clean data to fine-tune the moving mean and variance in BatchNorm. (don't update any parameters, just change the moving mean and variance value based on clean data). The experimental results are shown in the table:
>
> |Model     | 16k        | 32k           | 64k  |
> |:-------------| :-------------: |:-------------:| :----------------:|
> | **ResNet-50+PureAdv**      |  55.3   | 55.9   | 47.9   |
> | **ResNet-50+PureAdv+Finetune**      | 74.5   | 74.3  | 72.8  |
>
> From this table, we can find that the accuracy will be significantly improved after fine-tuning the moving mean and variance in BatchNorm. That means the BatchNorm is the main reason that the pure adversarial training cannot obtain a great performance.
>
> To further analyze the effects of BatchNorm, we try to concatenate the clean data and adversarial data and just use one shared BatchNorm (not split BatchNorm) to train the model. The experimental results are shown in the table:
>
> |Model     | 16k        | 32k           | 64k  |
> |:-------------| :-------------: |:-------------:| :----------------:|
> | **ResNet-50+PureAdv**      |  55.3   | 55.9   | 47.9   |
> | **ResNet-50+PureAdv+CleanData**      | 76.9   | 76.7  | 75.6  |
>
> We can find that the clean data can further improve the performance compared with pure adversarial training.
>
>
> [1] Cihang Xie, Mingxing Tan, Boqing Gong, Jiang Wang, Alan L. Yuille, Quoc V. Le. Adversarial examples improve image recognition. CVPR 2020

---

> > ### Author Response · Authors · 2021-12-04
> > **Looking forward to any other questions**
> >
> > Dear Reviewer M9ZZ,
> >
> > Thanks for your valuable review and comments again.
> >
> > If you have any other questions, please feel free to ask us.

---

### Official Review · Reviewer_nTet · 2021-11-03

**Correctness:** 4
**Technical Novelty And Significance:** 2
**Empirical Novelty And Significance:** 3
**Recommendation:** 5
**Confidence:** 4

**Main Review:**

My main concerns are the scalability of the proposed augmentation strategy and the gap between theorems and experiments.

Details:
1. The experiments mainly focus on ResNet-50/ImageNet training. How to set a large batch size for a new training problem? Whether 96k, 64k or 32k batch size is suitable for other problems?
2. The authors actually prove a non-asymptotic convergence rate and I also notice that LARS is used for the proposed methods in experiments. Such a gap may lead to many unknown problems. Furthermore, according to my experience, LARS is sensitive to the hyper-parameters and NaN often occurs for large batch training.


**Summary Of The Paper:**

This paper proposes a novel strategy which adding adversarial data for training to improve the performance of large batch training. Experiments also verify its efficiency.

**Summary Of The Review:**

Above all, the authors should improve the theory for a better understand of the adversarial data. It would be better if the authors propose the strategy on how to choose a suitable large batch size for a given problems.

---

> ### Author Response · Authors · 2021-11-21
> **Response to Reviewer nTet**
>
> Thanks for your inspiring comments.
>
> **[Experimental Results]** We try to expand the experimental results in Table 3 of the paper and scale up the batch size of EfficientNet-B2 Model to 64k. The experimental results are shown in the table:
>
>
>  |Model     | 1k        | 4k           | 8k  |  16k  |  32k  |  64k  |
> |:-------------| :-------------: |:-------------:| :-----:| :-----:| -----:| :-----:|
> | **EfficientNet-B2**      |  79.5  | 79.4  | 79.5  | 79.4  | 79.3  | 78.9  |
> | **EfficientNet-B2+AA**      | 80.0  | 80.0  | 79.8  | 79.5  | 79.4  | 79.1  |
> | **EfficientNet-B2+AA+ConAdv** | 80.4 | 80.4  | 80.4  | 80.4  | 80.2  | 80.0      |
>
> From this table, we can find that ConAdv can still help to scale up the batch size to 64k for EfficientNet-B2. For example, we can achieve 80.0\% accuracy for AA+ConAdv when batch size is 64k, which obtains higher accuracy compared with vanilla EfficientNet-B2 (79.5\%).
>
> **[Batch Size for New Task]** The main purpose of large-batch training is to use as many machines as possible to accelerate the DNN training. Therefore, batch size usually depends on the number of machines that you can use to train the model. Our method in this paper is to scale up the batch size and meanwhile maintain a competitive accuracy.
>
> **[Convergence]** Related work about adversarial learning can converge at a rate of $4\sigma \sqrt{\frac{L\Delta}{T}} + \frac{L_{\theta x}^{2}\lambda }{\mu}$ (based on [1]).  Due to using the staled weight to collect adversarial data, our method has an additional term $\frac{\tau M L_{x \theta}}{L \mu}$ in the convergence rate.
>
> **[Sensitity of LARS]** You et al. try to analyze the sensitivity of LARS for hyperparameters in Figure 2(b) of [2]
> and the experimental results illustrate that LARS is more stable for learning rate and warmup epochs compared with SGD+Momentum in large-batch training.  More specially, the performance of LARS can maintain the performance over a large number of hyperparameters, while the Momentum SGD cannot obtain good accuracy regardless of hyperparameter tuning. In this paper, we mainly focus on large batch setting, maybe your experience is from small-batch training.
>
> **[NaN in LARS]** As for the problem of NaN, this may because the gradient norm $\|g\|$ is 0 when  calculate the learning rate ratio ($\|w\|/\|g\|$). We can define  a  condition to solve this problem: the ratio is 1 when $\|g\|$ or $\|w\|$ is 0, otherwise the ratio is $\|w\|/\|g\|$. That is also the same as the official implementation in Flax [3].
>
> [1] Aman Sinha, Hongseok Namkoong, Riccardo Volpi, John Duchi. Certifiable distributional robustness with principled adversarial training. ICLR 2018.
>
> [2] You Yang, Wang Yuhui, Zhang Huan, Zhao Zhang, James Demmel, Cho-Jui Hsieh. The Limit of the Batch Size. arXiv preprint arXiv:2006.08517, 2020.
>
> [3] https://flax.readthedocs.io/en/latest/_modules/flax/optim/lars.html#LARS

---

> > ### Author Response · Authors · 2021-12-04
> > **Looking forward to any other questions**
> >
> > Dear Reviewer nTet,
> >
> > Thanks for your constructive review and comments again.
> >
> > If you have any other questions, please feel free to ask us.

---

### Official Review · Reviewer_Zv8z · 2021-11-06

**Correctness:** 3
**Technical Novelty And Significance:** 3
**Empirical Novelty And Significance:** 4
**Recommendation:** 8
**Confidence:** 3

**Main Review:**

## Strength

- Well-written. The motivation is clearly depicted and so is the thought process to the proposed solution.

- The proposed approach is simple yet novel and useful. As demonstrated in the experiments, using stale gradients can achieve a great speed up over the DisAdv baseline with no noticeable accuracy differences.

## Weaknesses

- It seems that the local machine has to support two parallel models. From what I understand, Fig 1b demonstrates the case where the local machine supports a stale model and a current model so both can be executed by different workers in parallel. If a model is so large that a single machine can't fit two models, then it creates extra communication overhead with other workers for communicating stale weights and adversarial examples. Long story short, using stale weights as a solution might lead to additional communication overhead that might be an issue depending on the setting of the training system.

**Summary Of The Paper:**

This paper presents a simple algorithm named ConAdv to incorporate adversarial training into the large-batch training setting such that one can further increase the batch size without harming too much accuracy while maintaining the high utilization of the hardware. The core idea is to use adversarial training to improve the accuracy for large batch training and at the same time use stale weights to allow parallel computation of the adversarial example and the normal gradient updates. With his simple yet novel approach, the paper has demonstrated good performance on ImageNet with batch size as large as 96k while maintaining the accuracy above MLPerf's 75.9.

**Summary Of The Review:**

Overall, I think this paper is well-written and it provides a novel, simple, yet useful algorithm for further increasing the batch size for large-batch training.

---

> ### Author Response · Authors · 2021-11-21
> **Response to Reviewer Zv8z**
>
> Thanks for your constructive comments.
>
> We agree with you that ConAdv may create extra communication overhead. The main overhead that ConAdv creates is to send adversarial data between different nodes. However, the size of adversarial data is much smaller than the model, especially when the model is large.
> Concretely, each node only processes 64 or 128 examples in our experiments, and the size of adversarial data on each node is thus about 7.4M (64\*115k) or 14.7M (128\*115k) since each image is about 115k (150GB/1.3M) in ImageNet-1k. In comparison, the total model size is 98M (ResNet-50), 171M (ResNet-101), 528M (VGG-16) or 632M (ViT-Huge). That means the additional communication overhead is small compared with the original communication cost.

---

### Decision · Program_Chairs · 2022-01-20

**Decision:**

Accept (Poster)

**Comment:**

Thank you for your submission to ICLR.  This paper presents a straightforward but reasonable approach to (slightly) improving the performance of large-batch training via adversarial training.  The basic approach is to apply (small epsilon) adversarial training, shown to help performance in small-batch settings, but accelerate the method using stale parameters to allow for parallel computation of the perturbations.  This speeds up adversarial training while improving performance, enough to enable it to be more effective than existing techniques for this large-batch setting.

The reviewers are not entirely in agreement about this paper, but I personally found the objections of the reviewer to be fairly generic, and not really addressing the core contributions of the paper.  However, I also felt that the overall contribution of this work seems somewhat incremental, using a not-particularly-unexpected result (that we can use stale gradients for this form of adversarial training) to achieve moderate speedup in what ultimately seems like one point in hyperparameter space.

That all being said, though, clearly the authors are working within standard benchmark frameworks, and "simple" algorithms here are indeed a positive rather than a negative.  So I am inclined to slightly recommend the paper for acceptance.